# Validation of the Polish Problematic Tinder Use Scale and Its Relationship with Safe Sex Behaviors

**DOI:** 10.3390/ijerph20053997

**Published:** 2023-02-23

**Authors:** Magdalena Liberacka-Dwojak, Yasser Khazaal, Monika Wiłkość-Dębczyńska, Daria Kukuła, Anna Chechłowska, Aleksandra Kozłowska, Nikola Przywitowska, Emilien Jeannot

**Affiliations:** 1Department of Psychology, Kazimierz Wielki University, 85-064 Bydgoszcz, Poland; 2Addiction Medicine, Lausanne University Hospital and Lausanne University, 1015 Lausanne, Switzerland; 3Institute of Global Health, Faculty of Medicine, Chemin de Mines 9, 1202 Geneva, Switzerland

**Keywords:** Tinder, validation scale, sex behaviors

## Abstract

Introduction: Online dating is a common phenomenon. The manageability and access of the application allows people to quickly reach many potential partners, which can increase risky sexual behaviors. The Problematic Tinder Use Scale (PTUS) was developed and validated in a Polish population by analyzing the reliability, validity, and factor structure of the responses given by Polish-speaking participants. Methods: Two samples of adult Tinder users were recruited online. The first study aimed to perform the reliability coefficient Cronbach’s, interrater analysis, exploratory, and confirmatory factor analysis. The second sample was recruited to investigate the factor structure by combining it with the Safe Sex Behavior Questionnaire (SSBQ). The sociodemographic data, such as hours of use and number of dates, were also investigated. Results: The Polish participants’ responses to the PTUS (sample 1: N = 271, sample 2: N = 162) revealed the one-factor structure of the tool. The reliability of the measurement was α = 0.80. The construct validity was confirmed. The results showed a significant, negative, and weak correlation between the PTUS and SSBQ scores and their subscales: risky sexual behaviors (r = −0.18), condom use (r = −0.22), and avoidance of body fluids (r = −0.17). Moreover, the number of partners met in the real world had a statistically significant, moderate relationship with the PTUS scores. Conclusions: The PTUS measurement is valid and reliable for the Polish population. The findings highlight the need for harm prevention strategies related to potentially addictive Tinder use, as well as the possible risky sexual behaviors associated with dating app use.

## 1. Introduction

Online dating is a common phenomenon that generates opportunities to form and maintain relationships [1]. Smartphone-based, geolocalization dating applications allow people to find potential romantic or sexual partners that might be difficult to find in real life [2]. According to Statista’s Digital Market Outlook, online dating services are expected to reach 413 million active users worldwide by the end of 2022 [3]. Tinder is perhaps the most well-known dating application (Tinder app) worldwide, and in Poland, it is the most downloaded dating app [4]. 

Tinder (tinder.com) is a dating software app that uses a geolocation system that offers dating partners the locations of nearby users. The manageability and access of the application allows people to quickly reach many potential partners. Furthermore, users can anonymously reject or like the profiles by swiping left or right, allowing for matches and direct communication between people if there is a mutual liking [5,6]. Such increased access to partners, anonymity, or searching for mates locally can contribute to difficulties in controlling app usage [6]. The trend of dating apps is widespread. Regardless of a person’s gender, age, sexual orientation, relationship status, education level, financial status, or personality traits, individuals use apps [7]. Individuals claim that the ease of search for partners makes Tinder engaging and interesting, and thus is possibly addictive [6]. 

Dating apps provide a quick and powerful reward since users can receive favorable social feedback, which increases with the time spent on Tinder. In addition, the number of potential partners could make it difficult to stop swiping [6]. In consideration of such concerns, Orosz et al. [6] assessed addictive Tinder use by Griffiths’ [8] six-component model. The model includes: (1) dominating role of Tinder in thinking and behavior (salience), (2) mood changes after Tinder use (mood modification), (3) the need to spend more time on Tinder (tolerance), (4) experiencing unpleasant feelings when being offline (withdrawal), (5) the disruption in relationships and other activities due to the Tinder use (conflict), and (6) after abstinence or control, a propensity to return to previous Tinder using practices (relapse). 

Almost every aspect of life has been significantly affected by the digital age, including dating, learning, and health. Additionally, their reliance on digital services has recently risen due to the mandatory social distancing brought on by the COVID-19 pandemic. People arrange meetings, jobs, and even dates online [9]. This phenomenon probably contributes to the increasing involvement of people in digital dating services and may partly change the way people experience sexuality, romances, and relationships [10] and possibly lead to problematic Tinder use among some individuals.

The literature implicates an association between dating applications and sexual risk behaviors [11,12]. The ease with which potential partners might be reached can increase risky sexual behavior, especially in the case of addictive Tinder use. These geosocial networking apps for smartphones have opened up new opportunities for finding sexual partners. As a result of apps such as Grindr or Tinder, users (especially young men who have sex with men) are in alarming danger of contracting HIV [13,14,15,16].

Tinder is the most popular dating app in Poland [4]. The aim of the present study was to assess and validate, to the best of our knowledge for the first time, the measurement of the Problematic Tinder Use Scale (PTUS) in Polish and to explore the relationship between the PTUS and the Safe Sex Behavior Questionnaire (SSBQ) scores. Two studies were carried out to develop and validate the PTUS responses in the Polish sample by analyzing their reliability, validity, and factor structure. Study 1 aimed to perform the reliability coefficients Cronbach’s α, interrater analysis, exploratory, and confirmatory factor analysis based on the maximum likelihood method. Study 2 was conducted to investigate the construct validity of the PTUS scores.

## 2. Method

### 2.1. PTUS Validation Process

The validation process was performed according to Brislin’s [17] procedure, which consists of four steps. First, the translation of the English version of the PTUS was conducted by five psychologists fluent in English. Subsequently, several Polish versions were analyzed, and one version was established and retranslated into the original language. The retranslation was performed by a native speaker. Finally, the version prepared by the native speaker was compared with the original version, and minor changes were made. 

Subsequently, a descriptive analysis of the characteristics of the PTUS responses was conducted. The dimensionality of the responses to the scale was examined via exploratory (EFA) and confirmatory (CFA) factor analysis. EFA was performed by using varimax rotation. CFA was based on the maximum likelihood method. The model was determined by the following indices: χ^2^, root mean square error of approximation (RMSEA), standardized root means square residual (SRMR), comparative fit index (CFI), the goodness-of-fit statistics (GFI), and the incremental fit index (IFI). The reliability analysis of the PTUS responses was conducted using Cronbach’s alpha (α). The Kappa statistic was performed to analyze the interrater agreement. Construct validity of the measurement was assessed by relationships between PTUS scores and SSBQ scores. The rho Spearman correlations coefficient and bootstrap method (N = 5000; 95% CI) were used to examine the relationships, leading to the conclusion that people who are addicted to dating applications are more prone to engage in risky sexual behaviors [11]. As mentioned before, two studies were performed to assess the validity of the PTUS responses in the Polish sample. The descriptive statistics, EFA, CFA, and reliability analyses were performed on the first sample. The CFA and rho Spearman correlations were conducted on the second sample. All analyses were performed using SPSS v27 and AMOS package v27. 

It was assumed that the PTUS Polish version would be considered reliable and valid if:

1. The obtained reliability coefficients exceed 0.7, which is the value that qualifies the tool for use in scientific research.

2. The results of the confirmatory analysis support the assumed factor structure of the tool. The model fit indices will be as presented: GFI > 0.9, IFI > 0.9, CFI > 0.9, RMSEA < 0.08, and SRMR < 0.08. The indices were chosen using guidelines for determining model fit [18].

3. The relationship between scores on the questionnaires is confirmed.

### 2.2. Measures

The following instruments were used for examination:

Polish version of the Problematic Tinder Use Scale (PTUS)

The scale contains six items based on Griffiths’ model of problematic use that measures components, such as salience, tolerance, mood modification, relapse, withdrawal, and conflict. Response options ranged from 1 (never) to 5 (always). Higher scores indicate more problematic Tinder use. The English version of the PTUS was used to perform Polish validation as it was primarily created in English and subsequently translated into Hungarian [6]. The reliability of PTUS in the original version was α = 0.83. 

Safe Sex Behavior Questionnaire [19] (SSBQ)

The questionnaire contains 24 items that measure the frequency of use of recommended practices that reduce the risk of exposure to, and transmission of, HIV and other sexually transmitted infections (STIs). Response options ranged from 1 (never) to 4 (always). The questionnaire measures five components: risky behaviors, assertiveness, condom use, avoidance of body fluids, and avoidance of anal sex. Higher scores indicate the safer sex behaviors. Risky sexual behaviors indicate involvement in behaviors, such as using drugs before sexual intercourse or engaging in sexual intercourse on a first date. The assertiveness subscale indicates the ability to insist on healthy sexual practices. The reliability of the SSBQ in the original version was α = 0.82. 

The sociodemographic data were also collected. The questions regarded gender, level of education, marital status, sexual orientation, and using the free or premium app version. Four additional items regarding Tinder usage style were asked. They referred to the weekly frequency of Tinder use, the number of partners who met in the real world, and statements about how much the individuals agreed that they used Tinder to search for a romantic or sexual partner. The sociodemographic questions were presented at the beginning of the questionnaire. All questions were required to be answered before moving on to the next item. However, participants were informed that they could resign from the study at any time. 

### 2.3. Recruitment Procedure and Ethics

The Bioethics Committee of Nicolaus Copernicus University approved the study protocol. The study was conducted using Internet surveys (Google Forms) among Polish Facebook users. Snowball sampling was used as a method of reaching respondents. Two separate samples (sample 1: N = 271, sample 2: N = 162) were recruited during the two recruitment waves, both using Facebook groups. The recruitment was performed between June and August 2022. The pragmatic sampling was performed in order to reach the greatest number of participants. All of them filled out the online, anonymous versions of the questionnaires. The data were collected using Google Forms. The participants received no monetary reimbursement. 

### 2.4. Study 1 

#### 2.4.1. Participants

The adaptation was carried out on a sample of 271 people: 186 women and 85 men (M = 25.06; SD = 5.37; aged 18 to 52). The basic characteristics of the samples are presented in Table 1. Most participants were highly educated (n = 155) and single (n = 202). In total, 200 participants stated that they were heterosexual. The majority of the respondents (n = 247) used the standard, free version of the application. Most of them claimed to be using Tinder a few times a week (n = 113). Only 27.31% of participants (n = 74) reported that they were looking for a sexual relationship on Tinder, in contrast to the majority of participants (n = 153) who said they were looking for a romantic partner. The mean number of partners who met in the real world in the last three months was 2.43.

#### 2.4.2. Results

(1) Descriptive statistics

The descriptive statistics are presented in Table 2. Moreover, the correlation between the responses to the six items of the PTUS was assessed and is presented in Table 3. To demonstrate a normal univariate distribution, values for skewness and kurtosis between −2 and +2 are regarded as acceptable [20]; therefore, the obtained data (see Table 2) revealed that questions 3 and 6 indicate the non-normal distribution. An item-total correlation (see Table 3) was performed to check if any item in the set was inconsistent with the averaged behavior of the other; a correlation value of less than 0.2 indicates the item is not discriminating well [21]. According to the data, all items were eligible for additional analysis.

#### 2.4.3. Factor Structure

(1) Exploratory factor analysis

The factor structure of the participants’ responses to the PTUS was conducted by EFA using the principal axis method. The varimax rotation was not performed due to the extraction of only one factor. Loading items above 0.5 were considered. The Kaiser criterion of including factors whose eigenvalue is higher than 1 was applied [22]. To show if EFA could be performed, the KMO and Bartlett tests were measured. KMO was 0.81 and the Bartlett was χ^2^ (15) = 549.81 (*p* < 0.001), which means that there are significant correlations between variables and that EFA could be performed. The EFA showed that a single factor explaining 52.92% of the total variance was the best solution. All items fulfilled this criterion (see Table 4). 

(2) Confirmatory factor analysis

A CFA based on the maximum likelihood method was applied to confirm the possible single-factor solution of the participants’ responses to the PTUS. The one-factor structure of the PTUS measurement had an acceptable model: χ^2^ (df = 4) = 12.38; *p* = 0.01; RMSEA = 0.08; SRMR = 0.04; CFI = 0.98; IFI = 0.98; TLI = 0.92. Figure 1 presents the factor structure of PTUS. Factor loadings for all items ranged from 0.6 to 1.47 (see Table 5). The modification indices suggested correlating errors 4 and 5. Factor loadings above 1.00 could suggest that there is a high degree of multicollinearity in the data [23]. Factor loadings reveal the variance explained by the variable on the specific factor. The factor extracts enough variation from the variable if the factor loadings are larger than 0.7, which was evidenced by the acquired data.

(3) Reliability analysis

Reliability statistics are shown in Table 6. Cronbach’s α was acceptable (0.80), which is above the desired threshold [24]. The discriminatory power of the responses to all PTUS questions is presented in Table 6. The analyses showed a high discriminatory power of all questions. Thus, it could be concluded that the reliability of the one-factor PTUS is satisfactory. The interrater analysis, which was also performed, showed substantial rater agreement (Kappa = 0.77, *p* = 0.01). 

### 2.5. Study 2

Study 2 was performed to validate the replicability of the one-factor structure of the PTUS measurement using CFA and to check the construct validity by correlating it with SSBQ scores.

#### 2.5.1. Participants

A total of 162 respondents (aged 18–52 years, M = 25.07, SD = 5.37) participated in the study. Of these, 115 were female (71%) and 47 (29%) were male. Among the respondents, 57.4% had higher education, 42.6% had secondary education, 73.5% were single, 24.1% were in an informal relationship, 2.5% claimed to be divorced, 71.6% were heterosexual, 16.7% bisexual, 6.2% homosexual, and 5.6% were other. The average number of partners met in the real world was 2.64 (M = 3.59). This study was conducted online among Facebook users.

#### 2.5.2. Results

(1) Validation of the one-factor structure of PTUS

The PTUS and SSBQ scales were used. Cronbach’s α of the participants’ responses to the PTUS was 0.86. To validate the one-factor structure of the responses to the scale, the CFA was performed. Indices were obtained as follows: χ^2^ (df = 4) = 12.27; *p* = 0.002; RMSEA = 0.08; SRMR = 0.04; CFI = 0.97; IFI = 0.90; TLI = 0.97. This indicates that the one-factor structure model fits well with the data in Polish samples. 

(2) Construct validity of PTUS

The next step was to test factor structure by correlating it with the SSBQ. As the Shapiro–Wilk coefficient indicated the non-normal distribution of the responses, the Spearman correlation coefficient and bootstrap method (N = 5000; 95% CI) were used (Table 7). There was a significant, negative, weak correlation between PTUS and SSBQ total score and its subscales: risky sexual behaviors, condom use, and avoidance of body fluids. Assertiveness was not significantly correlated with PTUS. The results confirmed a general relationship between engaging in problematic Tinder use, risky sexual behavior, and the construct validity of PTUS, so the factor structure of the tool could be established. 

(3) PTUS and sociodemographic variables

Table 8 presents the differences between the PTUS scores and sociodemographic variables. As all the variables indicated the non-normal distribution, the one-way ANOVA and Mann–Whitney U were used to assess the differences. The rho Spearman correlation was analyzed for the age and number of partners met in the real world. The results showed that almost none of the sociodemographic variables differed between the PTUS responses. However, the average time of weekly Tinder use significantly differentiated PTUS scores. People who used Tinder every day showed a higher score on the PTUS (M = 17.4), while individuals using Tinder less than a few times a month had the lowest scores (M = 11.7). Individuals who admitted that they were searching for a sexual partner indicated higher scores (M = 13.26) than those who did not (M = 11.10). Moreover, the number of partners met in the real world had a significant, moderate relation with the PTUS scores. 

## 3. Discussion

This study aimed to examine the psychometric properties of a Polish adaptation of the Problematic Tinder Use Scale. Consistent with the original version (Orosz et al., 2016), the one-factor structure of the six items of the Polish version of the PTUS was supported by EFA and CFA. The Polish version showed adequate reliability and construct validity. This was associated with the Safe Sex Behavior Questionnaire. In general, these results suggest that the adapted PTUS is a valid and reliable measure of problematic Tinder use suitable for use in the Polish population. Table 9 includes the Polish version of the scale.

Comparably to other studies, the findings implicate that the uncontrollable use of dating applications is related to a higher risk of engaging in risky sexual behaviors [25,26]. Furthermore, individuals who used Tinder every day and met more partners in the real world were more prone to problematic Tinder use. Sawyer et al. [26] indicate that people using dating applications were twice as likely to have unprotected sex. According to Orosz et al. [6], Tinder’s unique attributes, such as geolocalization systems that make it easier to find mates, rewarding value, ease of receiving positive feedback, and simplicity of making a profile, can contribute to the development of problematic behavior. Moreover, dating applications are nowadays mobile apps, which allow individuals to use them everywhere and whenever they want [26]. These elements may have an impact on the main problematic features of mood modification, salience, tolerance, and relapse [6,8]. 

Such results suggest that the use of Tinder simplifies the connection with sexual partners, and thus individuals using dating applications are more prone to risky sexual behaviors [26]. Researchers showed that people using dating apps claimed to have more recent and lifetime sexual partners, more unprotected sexual contacts, and a history of alcohol or drug use before or during sexual intercourse [15,27]. It could be consistent with the obtained result that individuals who use Tinder more often and meet more partners in the real world are more likely to report risky sexual behaviors. As dating apps simplify the finding of potential sexual partners, problematic Tinder use plays an important role as a guideline for risky sexual behaviors. Such results indicate the function of dating apps as hookup apps that allow sexual activities [27,28]. Moreover, the ease of searching for immediate interpersonal relations could make individuals more vulnerable and thus contribute to the greater risk of sexually transmitted diseases [28]. These findings imply that everyone should have access to potential interventions for sexual health information, especially adolescents and young adults, who use dating apps the most. Grindr, a dating app for gay men, introduced an effective way to reduce the spread of HIV by offering free access to HIV home test kits through ads and full-screen notifications on Grindr [28]. This indicates that dating apps can potentially contribute to promoting the sexual health of their users.

Additionally, no associations were found between sociodemographic factors and problematic Tinder use. This might imply that dating apps are becoming more widespread, regardless of residence, level of education, or marital status. According to recent studies, both men and women use dating apps in approximately equal numbers [29]; however, the number of men and women in the current study was not equal. Furthermore, it has been stated that these apps are gaining popularity across all age groups, including seniors 65 years and older [30]. Evidence shows that dating app users tend to have completed at least secondary education, which is consistent with obtained data as there were no respondents with lower education [31]. Additionally, nowadays everyone can now afford to have access to the internet, which is one of the fundamental needs of the modern world [32]. The fact that most dating apps are free to download makes them more accessible as they can be used on every mobile device [30].

This study has a few limitations. It was cross-sectional research. Outcome factors were self-reported. Future research could examine longitudinal data and differences in the number of sexual partners between dating app users and those who do not use internet applications. Moreover, the study included only one measure to assess construct validity. According to previous studies and a recent review [33], problematic Tinder use and probably other online dating activities could be moderated or modulated by a number of factors, such as motives for use [34], sexual desire, neuroticism or other personality related constructs, and insecure attachment [5,34]. Further studies on this scale may include some of the above mentioned constructs as well as other psychological and social dimensions to better understand the dynamic interplay between such dimensions and problematic Tinder use. Additionally, the current study used an online survey, which has its disadvantages. Various sampling issues could hamper the effectiveness of the survey, as only specific users seem to respond to online questionnaires. Thus, more research on multiple populations is needed. Moreover, there was a weak correlation between the responses to the PTUS and the SSBQ; therefore, further studies should be carried out to confirm the benefit of using the PTUS as a measure of potential addictive use of Tinder. However, the Polish version of the PTUS is a psychometric tool with good psychometric characteristics that may be useful in estimating the likelihood of developing problematic dating app usage. Given the prevalence of dating apps, problematic Tinder use may be more widespread, particularly among young individuals.

## 4. Conclusions

The PTUS in Polish is a valid and reliable indicator of problematic Tinder use. The one-factor structure of the tool was verified using EFA and CFA. These findings will enable researchers to apply this scale to the Polish population. The findings may also be important for professionals working with young adults who may be more aware of the potential sexual risks associated with problematic dating application use. 

## Figures and Tables

**Figure 1 ijerph-20-03997-f001:**
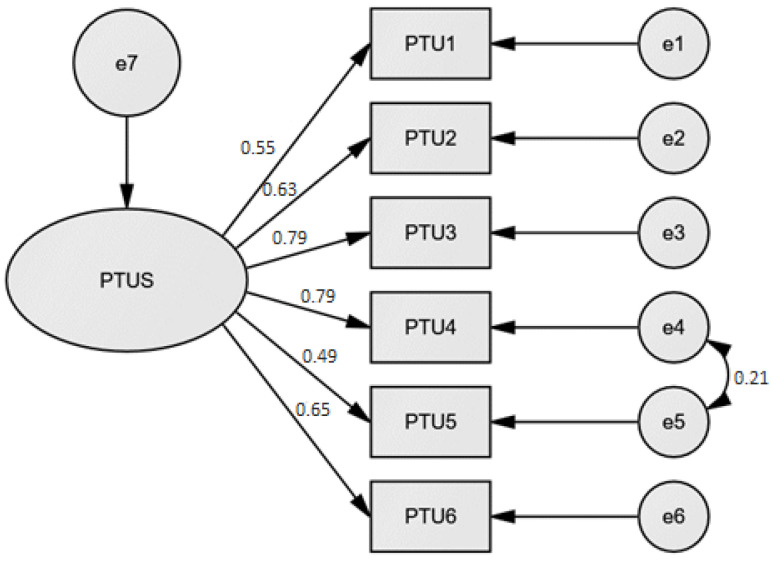
The factor structure of PTUS performed by CFA with standardized factor loadings.

**Table 1 ijerph-20-03997-t001:** Basic characteristics of the sample.

	n		%	
Sex				
Women	186		68.60	
Men	85		31.40	
Education level				
Secondary education	116		42.80	
Higher education	155		57.20	
Marital status				
Single	202		74.53	
Informal relationship	62		22.89	
Divorced	7		2.58	
Sexual orientation				
Heterosexual	200		73.80	
Homosexual	20		7.40	
Bisexual	39		14.40	
Other	12		4.40	
Tinder version				
Standard	247		91.10	
Premium	24		8.90	
Average time of weekly Tinder use				
Less than a few times a month	53		19.60	
A few times a month	31		11.40	
Once a week	29		10.70	
A few times a week	113		41.70	
Everyday	45		16.60	
Search for a romantic partner				
Yes	153		56.46	
No	49		18.08	
Do not know	69		25.56	
Search for a sexual partner				
Yes	74		27.31	
No	148		54.61	
Do not know	49		18.08	
	M	SD	min	max
Age	25.06	5.37	18	52
Number of partners met in the real world in the last three months	2.43	3.07	0	20

**Table 2 ijerph-20-03997-t002:** Descriptive statistics of PTUS (N = 271).

	M	SD	Skewness	Kurtosis
Thought about Tinder?	2.91	0.91	−0.03	−0.17
2 Spent much more time on Tinder than initially intended?	2.93	1.23	−0.03	−1.07
3 Become restless or troubled if you have been prohibited from Tinder use?	1.41	0.81	2.33	5.43
4 Deprioritized other hobbies and leisure activities because of your Tinder use?	1.55	0.96	1.68	1.73
5 Used Tinder in order to reduce feelings of guilt, anxiety, helplessness and depression?	1.99	1.25	0.92	−0.52
6 Tried to cut down on Tinder use without success?	1.38	0.83	2.39	5.29
PTUS total score	12.17	4.33	1.21	1.87

**Table 3 ijerph-20-03997-t003:** Correlation between items of PTUS (N = 271).

	PTUS1	PTUS2	PTUS3	PTUS4	PTUS5	PTUS6
Thought about Tinder?		0.57 **	0.34 **	0.36 **	0.26 **	0.23 **
2 Spent much more time on Tinder than initially intended?	0.57 **		0.42 **	0.45 **	0.29 **	0.41 **
3 Become restless or troubled if you have been prohibited from Tinder use?	0.34 **	0.42 **		0.61 **	0.37 **	0.49 **
4 Deprioritized other hobbies and leisure activities because of your Tinder use?	0.36 **	0.45 **	0.61 **		0.49 **	0.46 **
5 Used Tinder in order to reduce feelings of guilt, anxiety, helplessness and depression?	0.26 **	0.29 **	0.37 **	0.49 **		0.34 **
6 Tried to cut down on Tinder use without success?	0.23 **	0.41 **	0.49 **	0.46 **	0.34 **	
PTUS total score	0.68 **	0.81 **	0.63 **	0.69 **	0.67 **	0.57 **

** Correlation is significant, *p* < 0.01.

**Table 4 ijerph-20-03997-t004:** The exploratory factor analysis of PTUS (N = 271).

	Factor Loadings	Communalities	
		Initial	Extraction
Thought about Tinder?	0.66	1.00	0.44
2 Spent much more time on Tinder than initially intended?	0.73	1.00	0.53
3 Become restless or troubled if you have been prohibited from Tinder use?	0.79	1.00	0.64
4 Deprioritized other hobbies and leisure activities because of your Tinder use?	0.83	1.00	0.69
5 Used Tinder in order to reduce feelings of guilt, anxiety, helplessness and depression?	0.62	1.00	0.38
6 Tried to cut down on Tinder use without success?	0.71	1.00	0.50
Eigenvalue			3.18
Total variance explained			52.92

**Table 5 ijerph-20-03997-t005:** The CFA analysis results obtained from PTUS (N = 271).

Items	Factor Loadings	*p*-Value
Thought about Tinder?	1.00	0.001
2 Spent much more time on Tinder than initially intended?	1.54	0.001
3 Become restless or troubled if you have been prohibited from Tinder use?	1.28	0.001
4 Deprioritized other hobbies and leisure activities because of your Tinder use?	1.53	0.001
5 Used Tinder in order to reduce feelings of guilt, anxiety, helplessness and depression?	1.22	0.001
6 Tried to cut down on Tinder use without success?	1.08	0.001

**Table 6 ijerph-20-03997-t006:** The discriminatory power of PTUS questions.

Item No.	Discriminatory Power	Cronbach’s α, When Removed
1	0.53	0.78
2	0.58	0.77
3	0.65	0.76
4	0.70	0.74
5	0.46	0.80
6	0.54	0.78

**Table 7 ijerph-20-03997-t007:** Correlations between PTUS and SSBQ.

	Risky Sexual Behaviors	Assertiveness	Condom Use	Avoidance of Body Fluids	SSBQ Total Score
PTUS score	−0.18 *	−0.11	−0.22 **	−0.17 *	−0.27 ***

* *p* < 0.05; ** *p* < 0.01; *** *p* < 0.001.

**Table 8 ijerph-20-03997-t008:** The differences between PTUS scores and sociodemographic variables.

	F	MS	df	*p*-Value
Education level	0.561	0.364	2	0.43
Marital status	0.308	0.196	2	0.735
Sexual orientation	0.859	0.544	4	0.489
Average time of weekly Tinder use	15.217	7.083	4	<0.001
Search for a romantic partner	0.220	0.140	2	0.803
Search for a sexual partner	3.300	2.024	2	0.039
	U	Z		*p*
Sex	2498.50	0.75		0.45
	*r*			*p*
Age	0.009			0.91
Number of partners met in the real world in the last three months	0.37			<0.001

**Table 9 ijerph-20-03997-t009:** Polish version of PTUS.

Items
1. Myślał/myślała Pan/Pani o Tinderze?
2. Spędził/spędziła Pan/Pani na Tinderze więcej czasu niż początkowo Pan/Pani zamierzał/zamierzała?
3. Stawał/stawała się Pan/Pani niespokojny/niespokojna lub zmartwiony/zmartwiona jeśli nie mógł/mogła Pan/Pani korzystać z Tindera?
4. Odstawiał/odstawiała Pan/Pani na bok swoje inne hobby i sposoby spędzania czasu wolnego z uwagi na korzystanie z Tindera?
5. Odstawiał/odstawiała Pan/Pani na bok swoje inne hobby i sposoby spędzania czasu wolnego z uwagi na korzystanie z Tindera?
6. Bezskutecznie próbował/próbowała Pan/Pani ograniczyć korzystanie z Tindera?

## Data Availability

Data will be made available by the authors upon reasonable request.

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
