# Peer review of "Validation of the Polish Problematic Tinder Use Scale and Its Relationship with Safe Sex Behaviors"

_ijerph, 2023, doi:10.3390/ijerph20053997_

Round 1
Reviewer 1 Report
Validation of the Polish Problematic Tinder Use Scale and its Relation with Safe Sex Behaviors
I found the article informative, meticulously designed, and well-executed. The study provides a useful application of dating apps to monitor risky sexual behavior regarding the problematic tinder use scale (PTUS). The manuscript is well written, following are the specific comments to strengthen the manuscript further,
Specific comments,
1. Please consider changing relation to relationship in the title.
2. Line 71 and 73, in the introduction section the abbreviation form (PTUS) appeared first in line 71 and the full form later in line 73, consider swapping this, the full form Problematic Tinder Use Scale should appear first along with the abbreviated form (PTUS).
3. Concern about double the number of men in the study compared to women, while the statement of equal use of dating apps by men and women in line 263?
4. Line 269, Most dating also apps can be used on every smartphone and they are free to download, which makes them more accessible, please rewrite the sentence to make sense.
5. What recommendations are provided by the dating apps GRINDER to avoid HIV infection by young men who have sex with men?
Author Response
Thank you for your set of comments. We have tried to address the issue you have raised, as far as possible.
Comment 1: Please consider changing relation to relationship in the title.
Answer 1: We have modified the title, as suggested.
Comment 2: Line 71 and 73, in the introduction section the abbreviation form (PTUS) appeared first in line 71 and the full form later in line 73, consider swapping this, the full form Problematic Tinder Use Scale should appear first along with the abbreviated form (PTUS).
Answer 2: Thank you for pointing this out. We have incorporated your suggestion.
Comment 3: Concern about double the number of men in the study compared to women, while the statement of equal use of dating apps by men and women in line 263?
Answer 3: The statement of equal use of dating apps by men and women in line 263 referred to recent studies. In the current study, the number of participants was not equal which resulted from the recruitment process. We have modified the sentence in line 263 to avoid misunderstanding.
Comment 4: Line 269, Most dating also apps can be used on every smartphone and they are free to download, which makes them more accessible, please rewrite the sentence to make sense.
Answer 4: Thank you for the suggestion. We have rewritten the sentence.
Comment 5: What recommendations are provided by the dating apps GRINDER to avoid HIV infection by young men who have sex with men?
Answer 5: Thank you for the suggestion. We have added the recommendations provided by the Grindr to avoid HIV infection in lines 261-264.
Reviewer 2 Report
In the manuscript “Validation of the Polish Problematic Tinder Use Scale and its Relation with Safe Sex Behaviors”, Liberacka-Dwojak et al developed and validated The Problematic Tinder Use Scale (PTUS) for an online-dating app Tinder for finding potential partners. Two different studies were performed by analyzing its reliability, validity, and factor structure. Study 1 aimed to perform the reliability coefficients Cronbach’s α, exploratory, and confirmatory factor analysis based on the maximum likelihoods method. Study 2 was performed to investigate the construct validity of the PTUS. Tinder is the popular dating app, and the links between PTUS scores and the Safe Sex Behavior Questionnaire Scale (SSBQ) should be investigated. Based on the collective results, the authors concluded that PTUS is a valid and reliable tool for dating use for the Polish population. This paper is interesting and useful for further dating analysis. Accordingly, this reviewer recommends publication.
Author Response
Thank you for your set of comments. We appreciate your time and effort given to provide us the valuable feedback.
Reviewer 3 Report
See attached file.

Author Response
Thank you for your set of comments. We have tried to address the issue you have raised, as far as possible.
Comment 1: My major concern about the study deals with the
failure to include an adequate number of additional measures that would have provided better
data for construct validity. As it stands the authors only includes one measure (SSBQ) to assess
construct validity of reponses to the PTUS. The relations of the responses to the PTUS and the
responses to the subscales of the SSBQ were weak. Showing that responses to other construct
related measures and responses to the PTUS would have made this study more comprehensive.
Answer 1: Thank you for pointing this out. We agree with the reviewer, this is one of the study limitation. Accordingly, we acknowledged this study limitation in the discussion section as follows:
Moreover, the study included only one measure to assess construct validity. According to previous studies and a recent review [32], problematic Tinder use, like, probably, other online dating activities, could be moderated or modulated by a number of factors such as motives for use [33], sexual desire, neuroticism or other personality related constructs and insecure attachment [5,34]. Further studies on this scale may include some of the above mentioned constructs as well as other psychological and social dimensions to better understand the dynamic interplay between such dimensions and problematic Tinder use.
Comment 2: Page 1, line 16. "The Problematic Tinder Use Scale (PTUS) was developed..." Was developed to assess what?
Answer 2: Thank you for pointing this out. We have modified the sentence accordingly to your comment.
Comment 3: Page 1, line 17. "... by analyzing its reliability, validity and factor structure." Scales do not possess reliability, validity or factor structure. Participants' responses to the scale items possess such psychometric characteristics.
Answer 3: Thank you for this comment. We have modified the text.
Comment 4: Page 1 line 18. "...and validate PTUS. See the above comment.
Answer 4: Thank you. We have incorporated your suggestion.
Comment 5: Page 1, line 23. "...the one-factor structure of the tool." See the above comment.
Answer 5: Thank you. We have revised the text accordingly to your comment.
Comment 6: Page 1, line 23. "The reliability of the tool..." See the above comment.
Answer 6: Thank you. We have changed the phrase.
Comment 7: Page 1, line 28. "The PTUS is a valid and reliable tool..." See above comment.
Answer 7: Thank you. We have modified the text.
Comment 8: Page 2, line 71. "...to assess and validate the PTUS in Polish...". See above comment. Where is it noted that the original PTUS utilized the English language and that the psychometric characteristics of the responses to the scale items utilized what sample (were the participants Hungarians?).
Answer 8: Thank you for this comment. We have modified the text on page 2, line 71. The original PTUS was based on the wording of other questionnaires (Bergen Facebook Addiction Scale), and it was created in English and then translated into Hungarian according to the protocol of Beaton, Bombardier, Guillemin, and Ferraz (2000). The original English version was shown in the original article. The participants were Hungarians. We have changed the text on page 2, line 82 and added the comment about English-Hungarian translation on page 3, line 117.
Comment 9: Page 2, line 72. Why did the authors limit themselves in using only one scale for construct validity?
Answer 9: Thank you for pointing this out. We acknowledged the study limitation (see: Answer 1).
Comment 10: Page 2, line 73. "...validate the Problematic Tinder Use Scale...". See above comment.
Answer 10: Thank you. We have incorporated your suggestion.
Comment 11: Page 2, line 74. "...its reliability, validity and factor structure." See above comment.
Answer 11: Thank you. We have modified the text.
Comment 12: Page 2, Line 76. "...investigate the construct validity of the PTUS." See above comment.
Answer 12: Thank you for this comment. We have changed the phrase.
Comment 13: Going forward I will no longer address this issue. The authors should attend to this issue throughout the manuscript.
Answer 13: Thank you for such a detailed review. We have incorporated your suggestion throughout the manuscript.
Comment 14: Page 2, lines 79-84. Was the validation process quantified. Were there inter-rater agreement statistics?
Answer 14: Thank you for pointing this out. We have added the interrater agreement statistics (see: page 1, line 19, page 2, line 96; page 6, line 196).
Comment 15: Page 2, line 80. It is not clear whether the original form of the scale was in English or Hungarian?
Answer 15: Thank you. As mentioned above (see Answer 8, the original version was prepared in English).
Comment 16: Page 2, lines 102-104. Are there any standards for sample size when conducting CFAs? Did the authors obtain samples of sufficient size that met the standards?
Answer 16: Thank you for pointing this out. The pragmatic sampling was performed to reach the greatest number of participants (page 3, line 138). Nonetheless, the sample size is sufficient to meet the CFA standards (Studies show that a reasonable sample size for a simple CFA model is about N = 150).
See: Linda K. Muthén & Bengt O. Muthén (2002) How to Use a Monte Carlo Study to Decide on Sample Size and Determine Power, Structural Equation Modeling: A Multidisciplinary Journal, 9:4, 599-620).
Comment 17: Page 3, line 105. "....the relationship between scores on the questionnaires are confirmed." This is somewhat vague. If one is lookinig for construct validity data what kind of magnitude of the correlation would be expected (are there standard values for construct validity evidence?). Getting a weak relation between two scales would that argue for construct validity.
Answer 17: We appreciate your comment. Based on recent studies, the weak relation between the two scales was appropriate for confirming the construct validity.
See: Hagströmer, M., Oja, P., & Sjöström, M. (2006). The International Physical Activity Questionnaire (IPAQ): A study of concurrent and construct validity. Public Health Nutrition, 9(6), 755-762. doi:10.1079/PHN2005898
See: Koopmans, L., Bernaards, C. M., Hildebrandt, V. H., de Vet, H. C. W., & van der Beek, A. J. (2014). Construct Validity of the Individual Work Performance Questionnaire. Journal of Occupational and Environmental Medicine, 56(3), 331–337. https://www.jstor.org/stable/48500406
Comment 18: Page 3, lines 109-112. Reliability data is reported for the responses to the PTUS but no validity data are presented. is reliability from the current sample or from Gabor et al.? Also, why not also report Omega Squared which tends less biased when reporting Cronbach alpha for the current sample.
Answer 18: We appreciate your comment. Page 3, lines 109-112 refer to the original version of the PTUS (Orosz et al), see: page 3, line 117. The Measures section relied on the original versions of the scales. The Omega Squared was not reported in this section as Gabor et al. did not report this statistic. We hope that our explanation is appropriate.
Comment 19: Page 3, line 121. What about some validity data for the responses to the SSBQ.
Answer 19: We appreciate your time given to the comment, thank you. The reliability data of the original version of SSBQ is presented on page 3, line 126.
Comment 20: Page 3, Measures. What was the order of the various scales?
Answer 20: Thank you for your comment. The order was as presented in the Measures section: the sociodemographic data, PTUS, SSBQ. We appreciate your comment and have modified the Measure section to make it clearer what was the order of the various scales.
Comment 21: Page 3, Section 2.3. Were participants given the option not to respond to any item. Or were they forced to respond to each item before moving on to the next item?
Answer 21: Thank you for your comment. All questions were required to be answered before moving on to the next item. However, the participants were informed that they can resign from the study at any moment (page 3, line 134-135). The Bioethics Committee agreed to the proposed questionnaire form.
Comment 22: Page 3, line 132. Were there any Covid-19 protocols in effect during data collection?
Answer 22: Thank you for pointing this out. No Covid-19 protocols were provided as the survey was conducted online. Moreover, the survey was carried out in June-August 2022, while there were almost no Covid-19 restrictions in Poland.
Comment 23: Page 4, line 157. The data in the table represents 266 participants which indicates that 5 participants were excluded. Were they excluded due to missing responses? If so did the authors consider using multiple imputation techniques. Otherwise the Demographic data is not in line with the results for the PTUS.
Answer 23: We appreciate your detailed comment. We have checked the data and it appeared that showing 266 participants was our mistake. There were 271 participants. We have modified the text accordingly to this comment (page 4, line 124; page 5, line 165; page 5, line 177; page 6, line 191).
Comment 24: Page 4, 2.4.2 Results. What do we learn from the item means and standard deviations? Is the sample, in general, indicating problematic behaviors? Do these statistics line up with what has been reported in other studies using the PTUS?
Answer 24: Thank you for this comment. Accordingly to previous studies, the mean scores are similar to the one reported from other studies with convenience samples
(see: Orosz, G., Benyó, M., Berkes, B., Nikoletti, E., Gál, É., Tóth-Király, I., & Bőthe, B. (2018). The personality, motivational, and need-based background of problematic Tinder use. Journal of behavioral addictions, 7(2), 301-316.
see: Rochat, L., Bianchi-Demicheli, F., Aboujaoude, E., & Khazaal, Y. (2019). The psychology of “swiping”: A cluster analysis of the mobile dating app Tinder. Journal of Behavioral Addictions, 8(4), 804-813.).
Comment 25: Table 2. PTUS total score 2.03. This PTUS total score is the mean across the six items and not the total score. The total score would be 2.03 X 6 = 12.18. How does this total score compare to previous published findings (The Gabor et al. study reported a mean of 11.38). Need to know how comparable the current sample is to prior samples.
Answer 25: Thank you for this comment. We have modified the information in the table to make it comparable to previously published findings.
Comment 26: Page 6, Figure 1. Raw Factor loadings are presented. Orosz et al. used standardized factor loadings in their Figure. Might be beneficial to follow their lead so one can compare the two.
Answer 26: We appreciate your comment. We have changed the figure to show the standardized factor loadings, as presented by Orosz et al..
Comment 27: Page 7, Lines 218-227. It is not clear to me what specific statistical test was used to determine if the PTUS scores differed among the various demographics. Was a non-parametric test used? If so which specific one (was it the Mann-Whitney U)? Given that the total PTUS scores were within limits for skewness and kurtosis why resort to a non-parametric? Why was ANOVA ruled out? Were the variances of the groups equated (were the ANOVA assumptions met?)? Also, why was the rho Spearman correlation used? Were there outliers or lack of normality? Explain the rationale for the use of the Spearman.
Answer 27: Thank you for this comment. The One-way ANOVA and Mann-Whitney U were used to perform the differences between sociodemographic variables, as the Shapiro-Wilk coefficient showed that the data were non-normally disturbed. We have modified the text to show that (page 6, line 219; page 7, line 232).
Comment 28: Page 7, Table 8. I do not understand what the sd is referring to. If the Chi squared was used how does one also have a sd? This is confusing (unless I am missing something).
Answer 28: Thank you for your detailed comment. We revised the data and manuscript one more time and decided to change the statistics for the One-way ANOVA, as it was more appropriate to this dataset. We have modified table 8 accordingly to this suggestion.
Comment 29: Page 8, lines 271-277. Data was collected during the Covid-19 pandemic. Could this have influenced the results? What about social desirable responding? How was this addressed? What about the nature of recruitment? Could that have biased the results in any way (we know nothing about those who were recruited but refused to participate? Did not look at PTUS scores as a function of possible interactions among the demographics. The weak relation between the responses to the PTUS and responses to the SSBQ (and subscales) is problematic if one is attempting to show the benefit of using the PTUS as a measure of potential addictive use. More should have been done in establishing construct validity of responses to the PTUS.
Answer 29: Thank you for your suggestion. We have incorporated your suggestion and changed the text in the discussion (page 9, lines 293-299). As mentioned above, in Poland during the recruitment process, there were almost no further restrictions that could affect the results, so we decided not to mention it in the manuscript. We hope that this answer will satisfy you.